# On demand synthesis of hollow fullerene nanostructures

Fei Han[1,2], Ruoxu Wang[2], Yuhua Feng[1,2], Shaoyan Wang[2], Lingmei Liu[3], Xinghua Li[3], Yu Han [3] & Hongyu Chen[1,2]

Hollow nanostructures are widely used in chemistry, materials, bioscience, and medicine, but their fabrication remains a great challenge. In particular, there is no effective strategy for their assembly and interconnection. We bring pottery, the oldest and simplest method of fabricating hollow containers, into the nanoscale. By exploiting the liquid nature of the xylene template, fullerene hollow nanostructures of tailored shapes, such as bowls, bottles, and cucurbits, are readily synthesized. The liquid templates permit stepwise and versatile manipulation and hence, modular assembly of nodes and junctions leads to interconnected hollow systems. As a proof-of-concept, we create multi-compartment nano-containers, with different nanoparticles isolated in the separate pockets. This methodology expands the synthetic freedom for hollow nanostructures, building a bridge from isolated hollow units to interconnected hollow systems.

[1] Institute of Advanced Synthesis, Jiangsu National Synergetic Innovation Center for Advanced Materials, Nanjing Tech University, 211816 Nanjing, China. [2] Division of Chemistry and Biological Chemistry, School of Physical and Mathematical Sciences, Nanyang Technological University, 21 Nanyang Link, 637371 Singapore, Singapore. [3] Advanced Membranes and Porous Materials Center, Physical Sciences and Engineering Division, King Abdullah University of Science and Technology, Thuwal 23955-6900, Saudi Arabia. Correspondence and requests for materials should be addressed to H.C. (email: iashychen@njtech.edu.cn)

Pottery brings the dawn of manufacturing in human history. With it, one can readily design, build, and assemble hollow containers with shapes tailored for various applications. But this important concept has yet to be emulated in nanoscience. Hollow nanostructures have found applications in catalysis[1,2], nanoreactor[3], energy storage[4], and drug delivery[5]. Despite the progresses in application, the fundamental synthetic capabilities remain at a primitive level. Most hollow nanostructures are enclosed spheres[6–9] and to introduce an opening is still a great challenge[10–13]. Increase of structural complexity, for example, creating a nanoflask, is extremely difficult[14–17], not to mention more sophisticated shapes and systems.

Interconnected hollow systems are ubiquitous in modern life. Connecting hollow spaces can often bring about qualitative advance in property, for example, joining tubes into a channel system; connecting a condenser to a flask; and assembling space capsules into a station. However, such capability remains a blank at the nanoscale. To start on the problem, the elemental step would be connecting hollow nodes and junctions, which cannot be readily adapted from the existing methods for assembling nanoparticles, particularly considering the need to avoid barriers among the compartments.

We report a facile synthetic methodology: nanopottery, where hollow fullerene nanostructures with tailored opening and shapes can be readily created (Fig. 1). Importantly, the method allows flexible interconnection of hollow units, advancing from individual hollow units to multi-compartment vessels. The core synthetic capability comes from the stepwise manipulation of liquid template and the regio-selectivity allowed by the single opening, both of which are the focus of mechanistic study. As a proof of concept, we construct multi-compartment nanocontainers for the holding and isolation of different nanoparticles within one hollow system.

## Results

**Synthesis of $C_{60}$ nanobowls and bottles.** In a typical synthesis, $C_{60}$ solution in 1,3-dimethylbenzene (*m*-xylene) was added dropwise to *N, N*-dimethylformamide (DMF) with vortex, and 2-propanol (IPA) was then added to reduce the solubility of $C_{60}$. The solution changed gradually from nearly colorless to light yellow (Supplementary Fig. 1), indicating the slow nucleation of $C_{60}$. After 18 h, hollow $C_{60}$ hemispheres resembling bowls were obtained, with uniform size and opening (Fig. 2a–f and Supplementary Figs. 2, 3). When the rate of adding $C_{60}$ solution was increased (from 3 to 5, 10 and 20 μL/s), the half-shell changed towards complete sphere, with the size of their opening decreased from 181 to 102, 77, and 41 nm, respectively. When the $C_{60}$ solution was added in one shot, enclosed hollow spheres were obtained.

Interestingly, for nanobowls with small opening (e.g., 41 nm), prolonged incubation (>18 h) led to gradual outward extension of the $C_{60}$ shell, giving short tube-like protrusions (Fig. 2g, h), which eventually developed into long straight bottlenecks (119 ± 23 nm) at 108 h (Fig. 2i, j, m, p). With the bottleneck consistent in width with the nanobowl's opening, the overall structures resemble bottles or vases. During the growth, the bottle wall also increased in thickness (Supplementary Fig. 4D), indicating deposition of $C_{60}$ at its outer surface. The formation of bottleneck critically depends on the size of nanobowl's opening. No bottleneck was observed when the opening was over 77 nm, even after incubation for several days. Dynamic light scattering (DLS) measurements further confirm the formation of bottleneck in solution phase (Fig. 2o). It is possible that the critical factor is the small opening's ability to restrict the material flow from the inside of the nanobowls (vide infra). High-resolution transmission electron microscopy (HR-TEM) and selected area electron diffraction (SAED) results show that the $C_{60}$ shell in these hollow structures

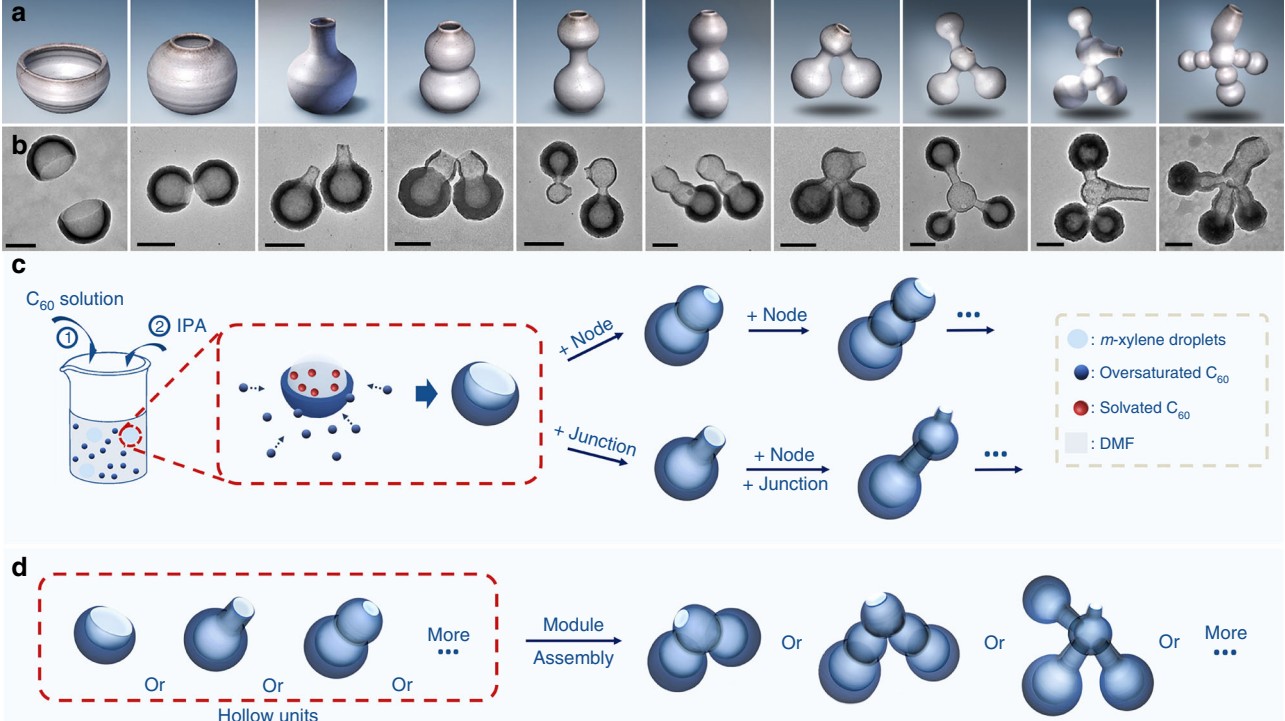

**Fig. 1** Schematics illustrating the concept and design. **a** Models of pottery, and **b** transmission electron microscopy (TEM) images of their corresponding nanostructures created with nanopottery. **c** Schematics illustrating the synthesis of different hollow units, and **d** connection of hollow units. Scale bars are 200 nm

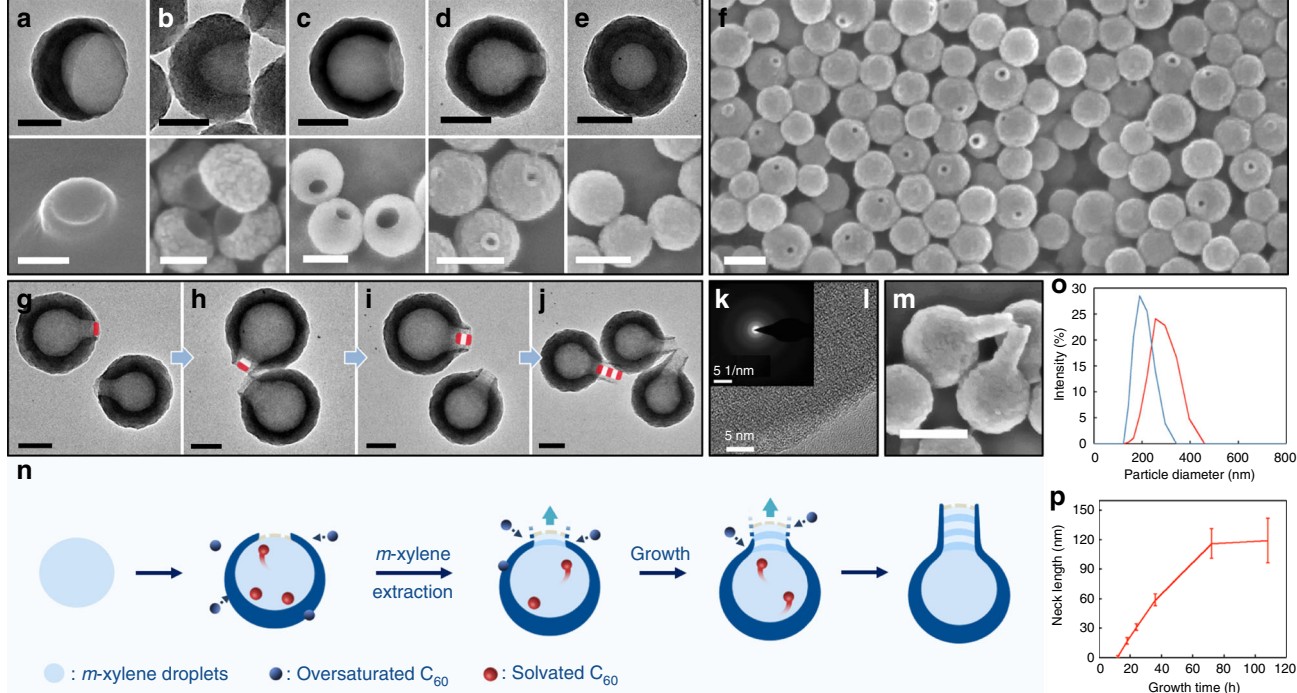

**Fig. 2** Morphological control of $C_{60}$ nanobowls and bottles. **a–e** TEM images (top row) of typical $C_{60}$ nanobowls with decreasing opening size and their corresponding scanning electron microscope (SEM) images (bottom row). **f** Large area SEM image of nanobowls with ~40 nm opening. **g–j** TEM images of nanobottles incubated for 18, 24, 36, and 72 h, respectively. **k, l** SAED and HR-TEM image of the nanobowl. **m** SEM image of nanobottles incubated for 72 h. **n** Schematics illustrating the formation of the nanobottles. **o** DLS measurement of particle size distribution of nanobottles incubated for 18 h (blue) and 72 h (red). **p** Dependence of nanobottles' neck length on incubation time; error bars indicating standard deviation. Scale bars are 100 nm for TEM images (the top row of **a–e**, and **g–j**), and 200 nm for SEM images (the bottom row of **a–e**, **f**, and **m**)

is amorphous (Fig. 2k, l). The as-prepared hollow structures have high stability: the nanobottles in water were not damaged after vigorously sonication for 1 h; the shape of the nanobottles remained unchanged after heat treatment at 80 °C for 24h; and the nanobowls could be stored in the original preparative solution for at least 3 months.

**Xylene droplet as template**. Considering the low solubility of $C_{60}$ in xylene/DMF/IPA, the amount of $C_{60}$ making up the bottleneck appears to be too much, given the maximum solute allowed in the small cavity of the initial nanobowls. Thus, we initially hypothesize that amorphous $C_{60}$ is the template, which eventually disappeared after ripening. However, DLS results showed instant formation of 100 nm droplets/particles after mixing $C_{60}$ solution in *m*-xylene with DMF (Supplementary Fig. 5A). When IPA was added, their size rapidly increased to over 120 nm. By the end of 150 min, the size increased to ~170 nm. The gradual growth of the particles argues against the solid template hypothesis. *M*-xylene is only partially miscible with DMF[18]. We ran a mock reaction by mixing *m*-xylene, DMF, and IPA according to the ratios used and in the absence of $C_{60}$. The mixture showed obvious Tyndall effect (Supplementary Fig. 6), and the DLS results confirmed the formation of droplets with diameter around 140 nm (Supplementary Fig. 5B). Hence, it is more likely that *m*-xylene droplets (containing some DMF and IPA) are the main template, whereas the $C_{60}$ precipitate in the droplets may provide the materials for the growth of bottlenecks.

Given the hypothesis, we attempt to extend the template, and thus the resulting hollow structure, by adding additional $C_{60}$ solution in *m*-xylene to the nanobowl solution. After 20 h, a second hollow compartment grew along the original nanobowl's

opening, forming a two-node, cucurbit-like structure (Fig. 3a–g). With further additions of $C_{60}$ solution, similar nodes can be sequentially added, giving interconnected hollow structures with exactly three, four, and five nodes (Fig. 3c–h). Such structures with exact number of connected hollow compartments are unprecedented. The arrangement of the nodes is roughly straight, with the opening often occurring at the far end. The wall thickness of the nodes decreases in sequence, with the newly grown one being the thinnest. Thus, there are two nucleation sites for $C_{60}$: one selectively along the opening extending the shell coverage, and the other non-selectively at the outer surface making all shells thicker.

**Tuning of the cavity size and bottleneck**. When the $C_{60}$ solution of the second addition was reduced in volume, the inner diameter of the second node changed accordingly (Fig. 3i and Supplementary Fig. 7). The correlation between them can be fitted with the following function (1):

$$y = (A\sqrt[3]{x} + B)10^{-7} \quad (A = 616.34, B = -79.44, R^2 = 0.98),$$
(1)

in which the cavity size ($y$) is proportional to the cubic root of xylene volume ($x$). This correlation gives a strong support that the nodes are templated by xylene droplets.

The second nodes have uniform size distribution, indicating that they were formed by gradual merging with a large number of tiny droplets, as opposed to random merging with only a few large droplets, which would lead to nodes with large size variation. We note that the bottleneck connects with the initial bowl with a smooth curve (Fig. 4h), whereas the cucurbit shows an abrupt turn at the connecting point (Fig. 4i). The former is

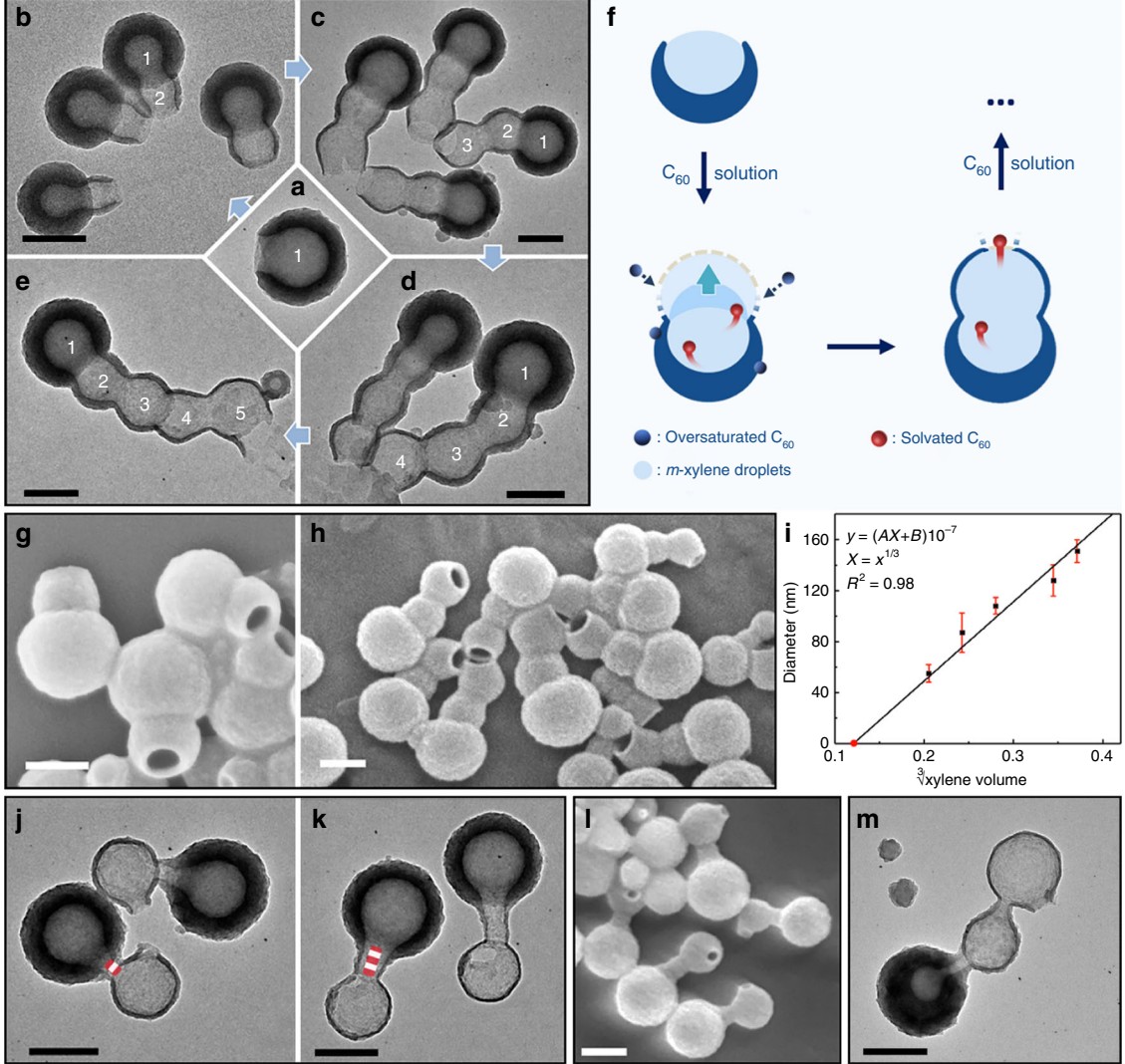

**Fig. 3** Addition of nodes and junctions to nanobowls and bottles. TEM images of **a** typical $C_{60}$ nanobowl, and **b–e** nanocucurbits with 2–5 nodes, respectively. **f** Schematics illustrating the formation of the node. **g, h** SEM images of nanocucurbits with two and three nodes. **i** Graph showing the linear relationship between the cavity diameter and the cubic root of the xylene volume. **j, k** TEM images of long-neck cucurbits with different neck length. **l** SEM image of long-neck cucurbits with neck length of 95 nm. **m** Addition of a node on long-neck cucurbits. Scale bars are 200 nm

consistent with a slow growth of the liquid domain, whereas the latter shows a rapid growth, that is, relative to the rate of shell formation. When the $C_{60}$ solution of the second addition was further reduced in volume, a much smaller and straight node that resembles a bottleneck was obtained, with only a smooth curve at the connecting point (Supplementary Fig. 7C). On these bases, we believe that the xylene solution is highly dispersed when mixed with DMF/IPA, and the resulting droplets gradually merge at the exposed liquid surface of the bowls/cucurbits. In contrast to the fresh addition of xylene droplets for cucurbits, the merging with the remaining xylene droplets at the later stage of growth is expected to be much slower. Such slow merging would cause the liquid to slowly exceed the rim of the nanobowls, forcing the $C_{60}$ shell to gradually extend outward, and form the straight bottlenecks (Fig. 2n). In other words, the bottleneck becomes straight when it is templated by many stages of thin layers, whereas a round node is formed when the liquid droplet grows much faster than the hardening of the $C_{60}$ shell (Figs. 2n vs. 3f).

To promote bottleneck formation, we attempted to introduce water into the system. It is expected to increase the polarity of the DMF/IPA phase, thus promoting the phase separation of xylene droplets and their subsequent merging (Supplementary Fig. 6C). When the DMF used for nanobottle synthesis was pre-doped with water (1–2% V/V), the resulting nanobottles have longer bottlenecks (160–190 nm) within a shorter period (36 h), matching with our expectations (Supplementary Fig. 8). It also provides a faster route to nanobottles.

**Study of the intermediates.** The growth intermediates of the nanobowls showed that thin half-shells formed at 2 h (outer diameter ~170 nm, Supplementary Fig. 9), and they grew thicker and gradually became more complete with reduced opening size. At the same time, the paste-like substance inside the shell, possibly xylene-solvated $C_{60}$, reduced and eventually disappeared. As the outer diameter of the half-shell increased, the diameter of its inner cavity remained almost unchanged (~140 nm), suggesting that the growth mostly occurs at its outer surface. It appears that the deposition of $C_{60}$ did not occur within the droplet, but has to be assisted by the outer solution. Based on this, we propose that the $C_{60}$ paste in the xylene phase may slowly dissolve into the IPA/DMF phase and re-deposit to give more stable $C_{60}$ domain.

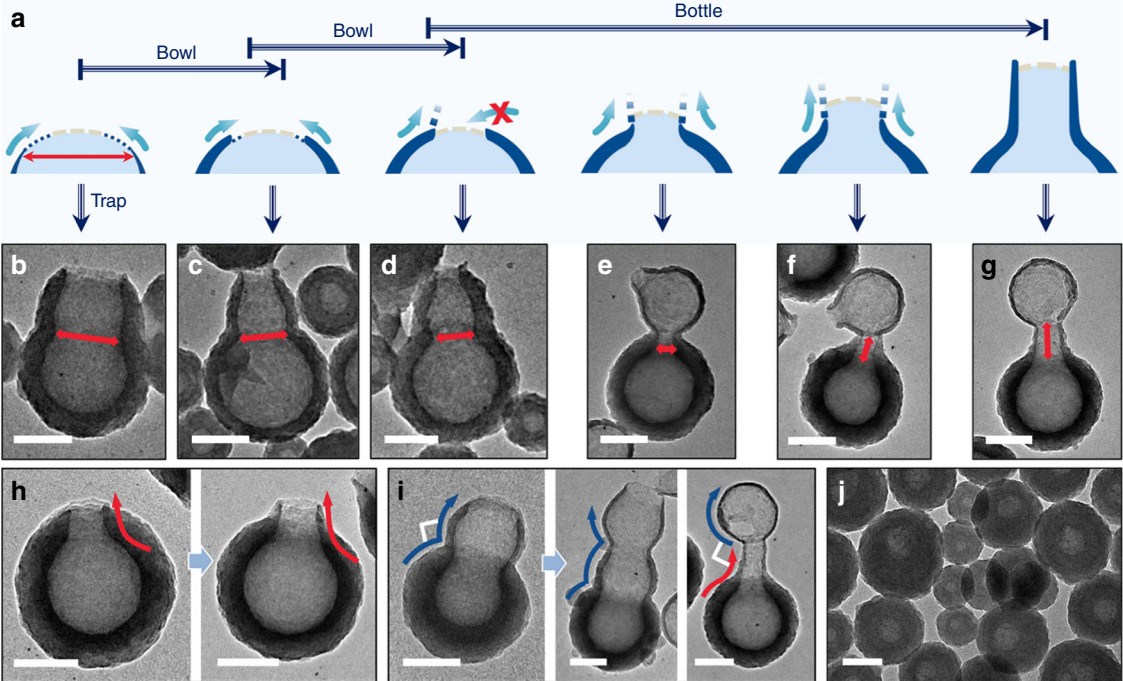

**Fig. 4** Trapping intermediates with the second addition. **a** Schematics illustrating the process of shell extension. TEM images of **b** second addition products of nanobowls reacted for 6 h; **c–e** second addition products of nanobottles reacted for 3, 6, and 15 h, respectively; **f, g** typical long-neck cucurbits with different neck length; **h** gradual curve at the connecting point for bottlenecks vs. **i** sharp turn at the connecting point for nodes growth, and **j** second addition products of enclosed spheres reacted for 3 h. Scale bars are 100 nm

This would allow $C_{60}$ in the paste to move into the ripening cycle, and its low solubility explains the long growth time of the bottlenecks.

Intermediates of the cucurbits showed paste-like substance at the nanobowl's opening at 1–3 h (Supplementary Fig. 10). Then, the new wall became recognizable, and slowly extended outwards from the opening, apparently along the curved edge of the second xylene droplet. As it grew thicker and larger, the new opening closed in and eventually a near-spherical node emerged (13 h). During this process, there was no barrier covering the original opening or the later junction between the two compartments. This result supports our hypothesis, as the new xylene domain is expected to quickly merge with the original droplet, forming a connected liquid phase (Fig. 3f). The transport of $C_{60}$ from the inside to the opening, and the material exchange with the outer solution keeps the opening open. Thus, it is conceivable that having an opening at the far end of the cucurbits would allow a straight and optimal diffusion path. This explains the coherent orientation of the openings in the cucurbits (Fig. 3b–e).

The selectivity for one opening per particle is extremely high. We did not find a single example with more than one opening. It is because that the growth of $C_{60}$ shell always starts from the rim of the opening and extends along the boundary with the opening reducing in size. Such gradual extension avoids multiple openings. In comparison to the heterogeneous nucleation at the rim of opening, the homogeneous nucleation at the droplet–solution interface would be less favorable.

We realize that the second growth is a convenient method for trapping the solution species, thus avoiding the inherent ambiguity from the drying process. As shown in Fig. 3, the second addition of $C_{60}$ solution in *m*-xylene to nanobowls gave additional nodes, but the opening at the cucurbit junction was kept as wide as the initial nanobowls. When the second addition occurred earlier at 6 h, as opposed to the typical 16–18 h, cucurbits with wider junctions were obtained (Fig. 4b and

Supplementary Fig. 11E). For nanobottles, when the second addition occurred at 3, 6, and 15 h, the trapped intermediates showed cucurbits with decreased opening size at the junction (from ~85 to 45 nm, Fig. 4c–e). Hence, there are clear trends that: (1) the opening decreases with time; and (2) the bottleneck only occurs after the opening is reduced to a critical size.

After the second growth on nanobottles, an additional node was found at the far end of the bottleneck, giving long-neck cucurbits (Fig. 3j–l). The selective addition at the bottleneck and the sharp turn at the connecting point supports the above mechanisms. The openings of the long-neck cucurbits have random orientation, unlike those without bottlenecks (Fig. 3b–d). The restriction by the narrow bottleneck is expected to disrupt the diffusion path of the material exchange, causing the opening to deviate from the straight path. Compare with the original nanobottles, their neck length remained unchanged after the node growth (50 and 100 nm). The bottleneck serves as a junction between the nodes, and additional junctions and nodes can be easily added by repeating the steps of neck growth and node growth (Figs. 3m, 4f, g). The level of structural precision achieved is amazing: not only are we able to control the exact number of hollow compartments but also the presence of junction and its length.

For comparison, second addition was added to the completely enclosed hollow $C_{60}$ spheres (same as Fig. 2e), only after 3 h of reaction. No extension occurred for those spheres, and instead, the excess $C_{60}$ formed small hollow spheres (Fig. 4j). This result proves the critical role of nanobowl's opening in the node growth: it not only allows the material exchange across the phases but also provides accessible area for the merging of droplets.

On the basis of the intermediates, we propose a coherent mechanism that is consistent with all observations: when $C_{60}$ in *m*-xylene solution was added into DMF, the reduced solubility excludes most xylene and $C_{60}$ from the DMF phase, forming tiny droplets that merge together. The addition of IPA further reduces

the solubility, allowing extensive growth until the exhaustion of $C_{60}$. The addition rate of $C_{60}$ solution determines the initial degree of $C_{60}$ oversaturation. As faster addition leads to more $C_{60}$ precipitate along the droplets, it determines the completeness or opening size of the initial thin shell. There are two nucleation sites: the non-selective growth at the outer surface depends on the $C_{60}$ concentration, which is a factor of the initial addition and the material exchange from inside of the xylene phase, minus the consumption by the growth. The selective growth at the rim of the opening is slow, as its collection area is much smaller than the outer surface. Hence, nanobowls with initial larger opening undergo faster material exchange and faster depletion of the $C_{60}$ paste. Only those with small opening could prolong the ripening process, so that their opening would be able to slowly close in and then form bottlenecks (Fig. 4a).

**Assembly and interconnection of hollow units.** In the TEM images, we note that the nanobowls, bottles, and cucurbits often arrange closely in a circle/curve with their openings pointing inward (Fig. 5a, b and Supplementary Fig. 2). This unusual alignment cannot be explained by random aggregation of the particles during drying. We speculate that they were assembled around a xylene droplet, which connects with the xylene domain inside the hollow units. This extra droplet may form at the initial stage of drying and eventually evaporated to leave a circular arrangement.

We attempted to cause $C_{60}$ growth around this hypothetical droplet. Water-doped DMF (1% V/V) was used to reduce the solubility of $C_{60}$, and the nanobowls used as the starting material were enriched to four times of the normal concentration, to enhance the probability of aggregation. After second addition of $C_{60}$ solution in $m$-xylene, interconnected structures were obtained. As shown in Fig. 5c, the dimers of nanobowls contain a shared hollow compartment as the node connecting the two nanobowls. When the nanobowls were replaced with nanobottles, connected trimers and tetramers were obtained in addition to dimers (Fig. 5g, h and Supplementary Fig. 12B), probably because the presence of bottlenecks reduced the mutual steric hindrance. When the nanobottles were not enriched, only dimers were formed (Fig. 5f),

agreeing with the concentration effect. In the products, the monomers are often well separated from each other surrounding the central node (Fig. 5g–j), with no direct contact between them. It is a clear proof that the monomers did not aggregate by direct contact, but via the mediating droplets at their openings.

Nanocucurbits could also be used as monomers for similar interconnection, giving a five-node structure with a single opening at the central node (Fig. 5e). Same as above, the size of the newly formed node could be adjusted by changing the amount of $C_{60}$ solution during the second addition (Fig. 5f, i). Moreover, when dimers of nanobowls were mixed with nanobottles, more complex multi-compartment vessels were obtained (Fig. 5d). These experiments provide proof of concept that various hollow structures could be interconnected, so long as their internal xylene domains could be linked via a new xylene droplet followed by $C_{60}$ over-growth (Fig. 5k). The multi-nodal structures are stable under normal centrifugation and transfer processes, or under heat treatment at 80 °C for at least 24 h. They could be stored for months in the original preparative solution (Supplementary Fig. 13B).

**Multi-compartment nanocontainers.** Due to the high stability of these hollow structures, as a proof of concept, we tried to construct multi-compartment nanocontainers with different substances held and isolated in the separate pockets. The different solvent environment in and outside the hollow structures provides an opportunity for selective loading of nanoparticles. Hydrophobic nanoparticles that can be well dispersed in xylene, for example, $Fe_3O_4$, Co, and Ag nanoparticles, were mixed with $C_{60}$ solution in $m$-xylene. They are expected to be trapped in the xylene droplets, and thus retained in the cavity. Figure 6a, e show Ag and $Fe_3O_4$ nanoparticles trapped in nanobowls, and Fig. 6b, c show Co and $Fe_3O_4$ nanoparticles in nanobottles. Elemental mapping of these containers confirmed the successful incorporation (Fig. 6f–l and Supplementary Fig. 14).

In contrast, when hydrophilic nanoparticles, for example, citrate-stabilized Au nanoparticles dispersed in DMF were used, they attached to the outer surface of the nanobottles (Fig. 6d), agreeing well with our synthetic design. Hence, during the

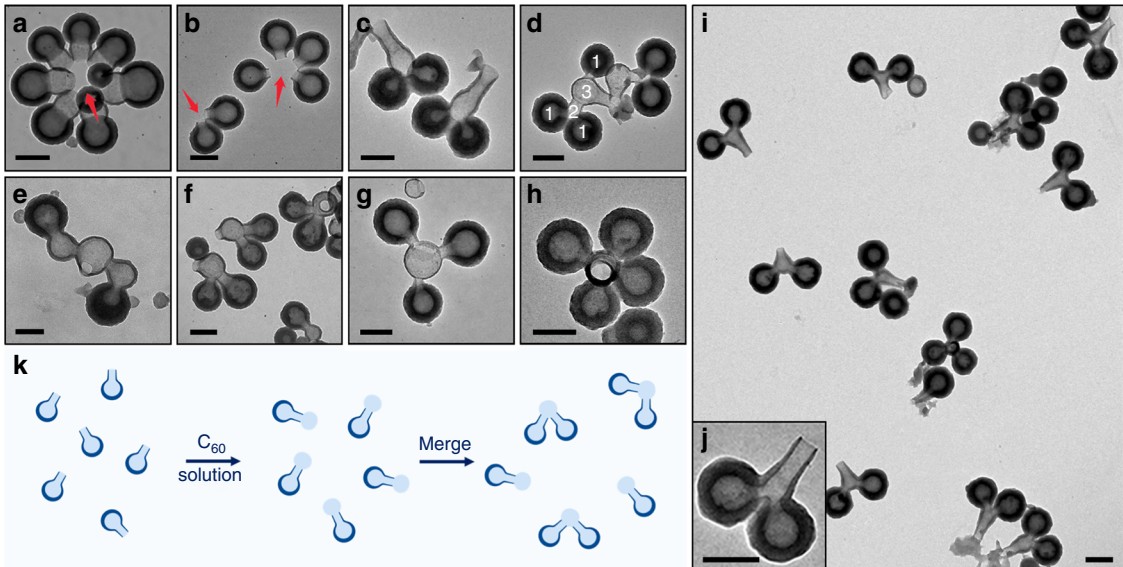

**Fig. 5** Connection of hollow units. TEM images of **a**, **b** $C_{60}$ nanocucurbits and bottles arranged in a circle; **c** dimers of connected nanobowls; **d** connection between a dimer of nanobowls and a nanobottle; **e** dimers of connected nanocucurbits; **f–h** dimers, trimers, and tetramers of nanobottles, respectively; **i** large area dimers of nanobottles, and **j** typical connected nanobottles. **k** Schematics illustrating the connection of nanobottles via the droplet at their opening. Scale bars are 200 nm

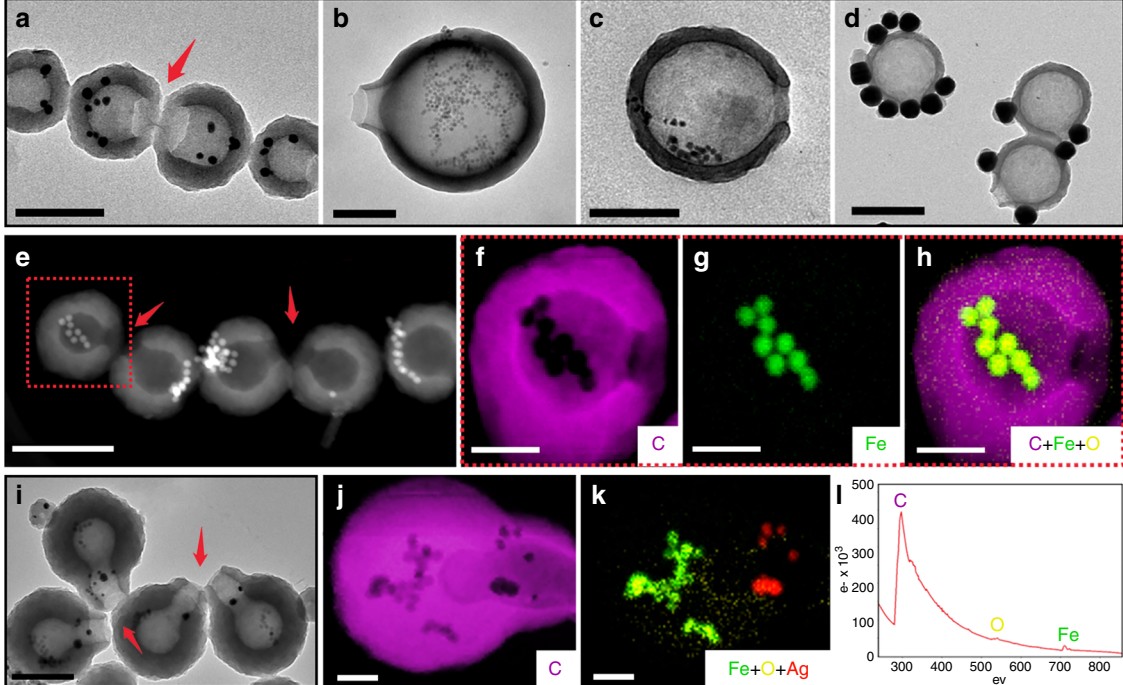

**Fig. 6** $C_{60}$ multi-compartment nanocontainers. TEM images of **a** nanobowls holding Ag nanoparticles; **b** nanobottles holding $Fe_3O_4$ nanoparticles; **c** nanobowls holding Co nanoparticles; **d** hydrophilic Au nanoparticles stay out of the nanobottles. **e** High-angle annular dark field (HAADF) image of nanobowls holding $Fe_3O_4$ nanoparticles, and its elemental mapping (**f–h**). **i** TEM image of multi-compartment nanocontainers with $Fe_3O_4$ nanoparticles in the bottom node and Ag nanoparticles in the upper node, and its elemental mapping (**j**, **k**). **l** Electron energy loss spectroscopy (EELs) of nanobowls holding $Fe_3O_4$ upper node. Scale bars are 200 nm for TEM images, and 50 nm for elemental mapping profiles

synthesis of cucurbits, when different nanoparticles were mixed in the multiple rounds addition of $C_{60}$ solution, they can be found in the separate compartments of the resulting cucurbits. For instance, Fig. 6i shows that the $Fe_3O_4$ nanoparticles are trapped in the bottom node, whereas the Ag nanoparticles in the upper node are consistent with their sequence of addition. Elemental mapping of this multi-compartment container further proves the successful incorporation and isolation of the nanoparticles in the separate compartments (Fig. 6j, k and Supplementary Fig. 15). The hydrophobic nanoparticles are expected to prefer the xylene phase than the DMF/IPA phase. The same could be said for any $C_{60}$ precipitated in our method. Hence, the selective positioning of the nanoparticles could support the selective localization of $C_{60}$ paste in the cavity, as discussed in the above mechanism.

**Generality**. The principles of nanopottery are expected to be generally applicable, though the conditions for different shell formation would need alternative design. Nanobowls could be created when the *m*-xylene solvent was changed to *o*-xylene, *p*-xylene, and toluene (Supplementary Fig. 16A–C). The resulting openings are shallower, likely due to the larger solubility of $C_{60}$ in these solvents. IPA could be replaced by ethanol or methanol, and nanobottles or hollow spheres were obtained (Supplementary Fig. 16D, E). In comparison, the modified solvent systems have similar reaction time, color change, and Tyndall effect, suggesting unchanged mechanism. In addition to $C_{60}$, other fullerene group materials like $C_{70}$ and PCBM (phenyl-$C_{61}$-butyric acid methyl ester) can also give similar hollow structures, by directly replacing $C_{60}$ in the reaction. The $C_{70}$ nanobowls can easily grow additional nodes, giving cucurbit structures (Supplementary Fig. 16F–H). The higher solubility of PCBM/$C_{70}$ requires higher concentrations and longer reaction time, but the general phenomena remain unchanged.

## Discussion

Fullerene and its derivatives have strong electron-accepting ability and excellent non-linear optical properties[19–21], being widely applied in lithium battery[22], solar cell[23–25], photo-conductive devices[21,26], and photocatalysis[20]. The new synthetic control provides precise and sophisticated structures with high surface area, opening new possibilities for further synthetic design.

In the conventional nanosynthesis, continual modification of a structure is extremely difficult. Most nanoparticles cannot be easily reshaped once the synthesis is completed. While liquid droplets can readily reshape by right, the colloidal droplets have monotonous spherical shape, due to the need to minimize their surface. Indeed, it has been almost impossible to manipulate the shape of nanodroplets. In contrast, this work demonstrates the dexterity of liquid droplets in shape control, and the use of them as template for a variety of hollow $C_{60}$ nanostructures. The material exchange between the *m*-xylene droplet and the external solution leaves a single opening that renders the critical regio-selectivity for subsequent modifications of the template, and thus hollow nanostructures. The liquid nature of the template allows continual structural modifications: gradual expansion of the xylene domain leads to a straight bottleneck, whereas direct addition leads to an extra spherical node. The processes can be repeated to give hollow structures with exact number of nodes, and bottleneck can be designed between any of the nodes. Merging the exposed xylene domains can connect bowls, bottles, or cucurbits together, forming multi-compartment vessels. Hydrophobic nanoparticles could be selectively retained in the cavities. Such stepwise expansion, addition, and connection are the basic skills of pottery, which would offer enormous synthetic freedom for designing complex hollow structures and inter-connected systems.

## Methods

**Materials.** Fullerene pristine powders ($C_{60}$, 98%), DMF (anhydrous, 99.8%), *m*-xylene (99%), and IPA (99%) are purchased from Sigma-Aldrich. All the chemicals were used without further purification. The experiments described could be reasonably reproduced.

**Synthesis of nanobowl/bottle/sphere.** $C_{60}$ powders were dissolved in *m*-xylene with a concentration of 0.80 mg/mL (for the synthesis using *o*-xylene as the solvent, the concentration is 2.2 mg/mL; for *p*-xylene and toluene as solvent, the concentration is 1.2 mg/mL; for the synthesis using PCBM/$C_{70}$ as shell material, the concentration is 14.5 and 2.4 mg/mL, respectively). Forty microliters of the above solution was added to 1 mL DMF dropwise (rate varies from 3 μL/s to one shot) with vortex, and 400 μL of IPA was added to the mixture slowly with vortex. The mixture was then sealed and incubated at room temperature. After 18 h, $C_{60}$ nanobowls were obtained, and $C_{60}$ nanobottles could be obtained after another 24 h.

**Synthesis of nanocucurbits.** One point two microliters of the nanobowl/bottle solution was taken, and 24 μL $C_{60}$ solution in *m*-xylene (0.80 mg/mL) was added quickly with vortex. The mixture was then sealed and incubated for 18 h at room temperature, giving the cucurbit structure. Following third, fourth, and fifth round addition of $C_{60}$ solution leads to cucurbits with 3–5 hollow nodes.

**Interconnection of hollow units.** Different individual hollow units were synthesized in water pre-doped DMF (1–2% V/V, without changing other steps). One point two milliliter of the above sample solution was taken and enriched to 1–4 times (via a 10 min centrifugation at 16$g$, and then remove certain volume of the supernatant). Twenty four microliters of $C_{60}$ solution in *m*-xylene (0.80 mg/mL) was added quickly with vortex. The mixture was then sealed and incubated for 14 h at room temperature, giving interconnected structures.

**Incorporation of metal nanoparticles.** Ag nanoparticles were synthesized using the literature method[27]. $Fe_3O_4$ nanoparticles were purchased from Ocean Nanotech. Co nanoparticles were purchased from Strem Chemicals. Solution of metal nanoparticles were mixed with $C_{60}$ solution in *m*-xylene (0.80 mg/mL) according to the ratio as shown in Supplementary Table 1. The mixture was used to synthesize nanobowl/bottle without changing the steps.

To synthesize multi-compartment containers, a second round of $C_{60}$ solution that contains another kind of metal nanoparticles was added ($V_{C_{60}+metal}/V_{nanobowl} = 1:40–1:48$).

**Fitting of the function.** The function between cavity size ($y$, in nm) and the xylene volume ($x$, in mL) was obtained using Microsoft Excel 2019.

**Characterization.** TEM images were obtained using a JEOL JEM-2100 electron microscope at an accelerating voltage of 100 kV. SEM images were obtained using a field-emission scanning electron microscopy (model JEOL 7600F) at an acceleration voltage of 5 kV. Prior to analysis, the samples were coated with gold layer using an Edwards Sputter Coater to enhance their conductivity. To prepare the sample, 0.4 mL of the sample solution was taken and centrifuged at the speed of 12$g$ for 8 min (for bowls, bottles, and cucurbits), or 1.5$g$ for 12 min (for more complex interconnected structures). The supernatant was removed, and the precipitate solution was dropped on a copper grid, which was pre-treated by a Harrick Plasma cleaner for 30 s to remove oxidations. For DLS measurement, 1 mL of the sample solution was injected to a four sides transparent glass cuvette, and then capped to avoid volatilize of solvent. A model ZEN 5600 from Malvern, and a back-scattering mode with an angle of 173° was used for all the measurements. SAED and HR-TEM were performed on a JEOL JEM-3010 TEM at 300 kV. HAADF-STEM imaging and EELs spectrum were carried out on a FEI-Titan ST electron microscope operated at 300 kV.

## Data availability

The data that support the findings of this study are available from the corresponding author on reasonable request.

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

## Acknowledgements

We would like to acknowledge funding support from the National Natural Science Foundation of China (No. 21673117), recruitment Program of Global Experts, Jiangsu Provincial Foundation for Specially Appointed Professor, start-up fund of Nanjing Tech University (39837102, 39837140), and SICAM Fellowship from Jiangsu National Synergetic Innovation Center for Advanced Materials. Ministry of Education Academic Research Fund (AcRF) Tier 1: RG9.12, RG10/16, RG111/15, Singapore. A*Star Science and Engineering Research Council – Public Sector Funding (PSF): 1421200075, Singapore and the National Research Foundation (NRF), Prime Minister's Office, Singapore under its Campus for Research Excellence and Technological Enterprise (CREATE) program.

## Author contributions

H.C. designed this project and proposed the droplet growth mechanism. F.H. synthesized all the hollow structures and helped in revising the mechanism. Y.F. and R.W. helped in revising the mechanism. S.W. helped in the SEM characterization. L.L., X.L., and Y.H. helped in the HR-TEM and EELs characterization.

## Additional information

**Competing interests:** The authors declare no competing interests.

