## [Peer Review File · Nature Communications]

Reviewers' comments:

Reviewer #1 (Remarks to the Author):

This manuscript by Chen et al. reports a collection of hollow C60 nanostructures by using liquid droplets as soft templates. The soft template allows versatile structure modifications. The images are impressive and eye-catching. However, my impression of this manuscript is that it presents a superficial look at a quite special piece of hollow structures, without providing enough discussion to understand what the underlying chemistry and to suggest what we can learn from it. At this point I cannot recommend publication in Nature Communications. The authors should think about the following points to improve the strength of their work:

The major result of the paper is using liquid droplets as templates for the assembly C60 into hollow structures. It is unclear to me why the authors chose C60? For example, in the manuscript there is no mention about why C60 hollow structures are unique and interesting? Can the strategy be extended to other materials system? or only works for C60? If it is applicable for other materials, especially for inorganic materials, it would be a breakthrough in the field of hollow nanostructure synthesis. Otherwise, it would not attract the community's much interests. I would suggest the authors to put more efforts in the introduction part to stress either the unique of synthetic methodology or the materials/structures.

In terms of presentation, the main issue is that it is too descriptive. For example, the organization of the manuscript is on the basis of the experimental process and still more a collection of facts; the so-called mechanism study is actually the description of TEM results of time-dependent morphology evolution (this is also true for the schematics of the formation of hollow structures). A weakness is a lack of understanding what the underlying chemistry looks like.

In the third part, the authors tried to trap nanoparticles with different surface chemistry into different parts of the hollow structures. This seems like interesting, given the controllable positioning of nanoparticles, but again, the presentation is quite descriptive. What do we learn from the positioning of nanoparticles? It would be good if the authors can take one more step, e.g., demonstrate the significance of this positioning of nanoparticles or a specific application that otherwise cannot be easily achieved in other system.

Reviewer #2 (Remarks to the Author):

This work reports the synthesis of a series of very delicate hollow C60 nanostructures. The authors demonstrated the dexterity of liquid droplets in shape control, and the use of them as template for a variety of very unique structures, including bowls, bottles, or cucurbits and multi-compartment vessels constructed from these structural elements, which were not possible to be prepared using conventional methods. The authors demonstrate elegant use of droplets for creating openings, extended bottlenecks, and interconnection. The interconnection is a new capability that will be obviously useful. While a few examples of similar bottlenecked structures are known, the previous syntheses are not well controlled to the same level. The authors provide detailed mechanistic analysis which is self-consistent. This work represents a significant progress in the design and synthesis of hollow nanostructures, and opens new pave towards potential new applications of hollow nanostructures. I would like to recommend acceptance of this manuscript after address the following minor issues.

1. The majority of works on hollow nanostructures produced spherical core-shell structures. In this work the authors demonstrated the preparation of bowl-like structure with well controlled opening. The mechanism of for the formation of such bowl structure should be explained more clearly.

2. Herein the authors fabricated the nanostructure using fullerene. The reason of choosing fullerene rather than other materials should be addressed. Potential applications of these nanostructures utilizing the unique optoelectronic properties of fullerene should be discussed.

Reviewer #3 (Remarks to the Author):

Fei Han et al. reported a very interesting method to synthesize hollow-structured fullerene. Hollow nanostructures have broad potential applications in nanoreactor, catalysis, energy storage, and so on. However, the fabrication of hollow nanostructures of tailored shapes on nano-scale remains significant challenges. In this work, the author reported a facile synthetic methodology of nanopottery, which could create different hollow units and connection of hollow units. The key innovation of this study is to control the shape of liquid droplet and then use them as template for a variety of hollow C₆₀ nanostructures. In fact, the liquid droplets were always spherical to minimize the surface tension in most system which was difficult to manipulate their shape. As such, the quality of the experimental data is superb and the manuscript is well written. The idea is novel and the experiments were well designed and conducted to prove the concept. If the fundamental mechanism and design principle could be more comprehensively understood, it would provide broader impact. I would recommend this work to be published if following concerns could be addressed:

1. All the nanostructures created with nanopottery were characterized by TEM. Could the author provide more characterizations with statistics significance? For example, the dynamic light scattering (DLS) can not only provide statistical information in the size but also in the structure factor of different hollow structures. How about the size and structure changes of nanobowls and nanobottles with various nodes or junctions?

2. What is the driving force for the C₆₀ in xylene to aggregate together and form the shell of the structure?

3. The authors only mentioned the influence of the rate of xylene on the structure. How about the influence of the rate of IPA?

4. Could the authors explain how the fullerene assembles into different structures? What is the binding morphology between the C₆₀?

5. According to the HR-TEM and SAED results, the C₆₀ shell in these nanobowls is amorphous. Are these nanocontainers durable in all pH environments? Can they stand heating? How about the chemical stability of nanobowls or nanobottles with various nodes or junctions?

6. The Figure 4F in the supplementary is missing.

7. From the SEM images showed in Figure 2 and 3, the surface of the nanoparticle is not smooth. What is the reason for this?

8. The authors used xylene, DMF and IPA as the solvents for all the experiments. How about other organic solvents, such as ethanol, methanol, THF, CHCl₃, acetone and DMSO?

9. In Figure 4i, the authors showed the connection of the hollow units. Is it possible to separate the dimer and trimer?

10. To broader application of this method, how to remove the excess C60 in the solution after the synthesis? In the present manuscript, the authors did not mention the purification process. Also, is the product with different nanostructures stable and easy to store? The authors are invited to add some more detailed information about this in the revised manuscript.

11. How to apply this material is an important issue. Could the authors add experiments to illustrate some potential application, for example, the possibility of application the nanoparticles contained in the nanobottles.

Response to Reviewers' Comments

Reviewer1:

Comments to the Author:

This manuscript by Chen *et al.* reports a collection of hollow C₆₀ nanostructures by using liquid droplets as soft templates. The soft template allows versatile structure modifications. The images are impressive and eye-catching. However, my impression of this manuscript is that it presents a superficial look at a quite special piece of hollow structures, without providing enough discussion to understand what the underlying chemistry and to suggest what we can learn from it. At this point I cannot recommend publication in Nature Communications. The authors should think about the following points to improve the strength of their work:

The major result of the paper is using liquid droplets as templates for the assembly C₆₀ into hollow structures. It is unclear to me why the authors chose C₆₀? For example, in the manuscript there is no mention about why C₆₀ hollow structures are unique and interesting? Can the strategy be extended to other materials system? or only works for C₆₀? If it is applicable for other materials, especially for inorganic materials, it would be a breakthrough in the field of hollow nanostructure synthesis. Otherwise, it would not attract the community's much interests. I would suggest the authors to put more efforts in the introduction part to stress either the unique of synthetic methodology or the materials/structures.

Response:

We thank the reviewer for the critical and valuable suggestions. We have added detailed discussion to provide deeper understanding of the reported phenomena. In the initial version, those were left out to minimize confusion. We do agree with the reviewer that the initial manuscript appeared too simplistic.

We would like to invite the reviewer to see our work from a different perspective, namely the development of a new capability, rather than the creation of a new structure/material. Just like in organic chemistry, their relative importance is self-evident: there are millions of organic molecules, but just so many types of organic reactions.

In organic chemistry, the discovery of new molecules is often not interesting per se; the breakthrough is the new capability or synthetic pathway demonstrated in making these molecules. For example, the Nobel prizes of olefin metathesis or Suzuki coupling are given for the development of new capabilities (of joining C-C bonds), rather than the initial forms of molecules per se.

As laid out in our introduction, we believe that the interconnection of hollow structures is a new capability, which at the macroscale has been shown to be extremely useful. The multi-compartment cucurbits with exact number of nodes, and the connection of bottles with different size/shape are demonstrations of the new capability.

In nanoscience, the capabilities of creating cubes, rods, hollow structures, *etc.*, at the time of their initial discovery, have inspired lots of interests in the community, and indeed, many discoveries followed since. For such initial reports (gold nanorods, silver nanowires, silica hollow structures, to name but a few), it appears to us that the most urgent issue is often not to establish the generality of the new capability, but the reason why such capability can be created.

C₆₀ is the first system where the new synthetic control is established. It is a popular material used for optical devices and solar cell. Among all molecular nanocrystals and nanoparticles, C₆₀ is probably the one with richest morphologies. Like Au nanostructures are intensely studied as the model system for metal nanostructures, C₆₀ is often taken as the model system for molecular nanostructures.

The mechanism of using liquid droplets for continual extension of hollow compartments (in the making of cucurbits and bottles) and for joining bowls, bottles, and cucurbits appears to be general. At least from the general principles we have established, there is no reason they cannot be applied to other materials, though the exact conditions may need some adjustment.

We have indeed explored the generality of this synthetic control. In addition to C₆₀, C₇₀ and PCBM (phenyl-C₆₁-butyric acid methyl ester) can also give similar hollow structures. Moreover, the method can be extended to inorganic materials: hollow silica cucurbits and iron oxide bowls/bottles were generated (Fig. R1). We have now included the C₇₀ and PCBM data in the supporting information (Supplementary Fig. S18). The inorganic material work involves completely different reaction conditions and consequences, and was kept out of the current manuscript to avoid confusion.

[Redacted]

Comments to the Author:

In terms of presentation, the main issue is that it is too descriptive. For example, the organization of the manuscript is on the basis of the experimental process and still more a collection of facts; the so-called mechanism study is actually the description of TEM results of time-dependent morphology evolution (this is also true for the schematics of the formation of hollow structures). A weakness is a lack of understanding what the underlying chemistry looks like.

Response:

We indeed agree with the reviewer and thank the reviewer for the valuable comments. We have made major revisions, to include more control experiments and detailed analysis.

We are trying to establish a coherent, self-consistent mechanism from multiple results, as opposed to hand-waving arguments that can only be applied to a single detail of the system. As proposed, our mechanism is consistent with and supported by the following key phenomena:

- 1) The hollow structures are templated via *m*-xylene liquid droplets. It is supported by the solvents miscibility data from the literature, by the mock reaction of mixing solvents, and by the matching size of droplets measured from DLS.
- 2) With the expanding of xylene droplets, individual hollow structures can be extended to become more complex structures. For instance, fast expansion of xylene droplets leads to a new node, and slow merging with the remaining droplets leads to a straight bottleneck. And further addition of xylene gives an additional node on top of the bottleneck. The mechanism is supported by the intermediates of bowls and cucurbits, by the control experiments of the second addition, and by the comparison of the detailed structure features among different hollow structures (sharp and smooth turning points).
- 3) The volume of the C₆₀ solution of the second addition, and the cavity size of the resulting node show a strong correlation, which is consistent with the hypothesis of liquid template.
- 4) The process of closing the opening was trapped by drying the growth solution. Using 2nd growth as an unusual means of trapping solution species (as opposed to drying), we showed the gradual closing of the opening and extension of the bottleneck, where all details are consistent with the proposed mechanism.
- 5) The arrangement of hollow nanostructures in a circular form on the TEM grid suggests possible templating liquid that has evaporated during sample preparation. Our method of interconnection was indeed developed by exploiting this phenomenon. Hence, our success strongly supports the liquid templating mechanism.

I would argue that, different from molecular-scale reactions, at nanoscale, the most important task of a mechanistic study is to understand which species plays what role at which time point, in a way similar to the mechanisms of cellular chemistry (Fig. R2). At this length scale, one would hardly discover new chemistry. Nonetheless, such knowledge is fundamental for achieving synthetic control and designing more sophisticated structures and functions.

Fig. R2. At different length scale, the meaning of mechanism is different.

Comments to the Author:

In the third part, the authors tried to trap nanoparticles with different surface chemistry into different parts of the hollow structures. This seems like interesting, given the controllable positioning of

nanoparticles, but again, the presentation is quite descriptive. What do we learn from the positioning of nanoparticles? It would be good if the authors can take one more step, *e.g.*, demonstrate the significance of this positioning of nanoparticles or a specific application that otherwise cannot be easily achieved demonstrate the significance of this positioning of nanoparticles or a specific application that otherwise cannot be easily achieved in other system.

Response:

We thank the reviewer for the critical and valuable suggestions.

With the synthesis of multi-compartment nanocontainers and understanding of the underlying mechanism, we want to demonstrate that nanoparticles could be selectively incorporated, as an extension of the nanocontainer concept.

Indeed, in nanoscience, it remains a great challenge to demonstrate the usefulness of sophisticated structures. We attempted to give an example of how to use the multi-compartment vessels to holding and isolating different nanoparticles. Such site-selective incorporation is a synthetic advance. It is a fundamental step towards future applications. In terms of the potential applications of these structures, we have added a corresponding discussion. We hope this synthetic advance could inspire new potentials for future applications.

In the history of mankind, simple metal structures such as sword and axe are useful, and so are the very complex machines and engines. But because metal structures of intermediate complexity are not very useful, it took mankind thousands of years to accumulate the knowledge and finally achieve the synthetic control. Should people focus on the development of new methods and capabilities, the technological leap would occur much earlier.

Reviewer2:

Comments to the Author:

This work reports the synthesis of a series of very delicate hollow C₆₀ nanostructures. The authors demonstrated the dexterity of liquid droplets in shape control, and the use of them as template for a variety of very unique structures, including bowls, bottles, or cucurbits and multi-compartment vessels constructed from these structural elements, which were not possible to be prepared using conventional methods. The authors demonstrate elegant use of droplets for creating openings, extended bottlenecks, and interconnection. The interconnection is a new capability that will be obviously useful. While a few examples of similar bottlenecked structures are known, the previous syntheses are not well controlled to the same level. The authors provide detailed mechanistic analysis which is self-consistent. This work represents a significant progress in the design and synthesis of hollow nanostructures, and opens new pave towards potential new applications of hollow nanostructures. I would like to recommend acceptance of this manuscript after address the following minor issues.

Response: We thank the reviewer for the high recognition of our synthetic advance.

Comments to the Author:

1. The majority of works on hollow nanostructures produced spherical core-shell structures. In this work the authors demonstrated the preparation of bowl-like structure with well controlled opening. The mechanism for the formation of such bowl structure should be explained more clearly.

Response: We thank the reviewer for the critical question.

In our revised manuscript, we provided detailed discussion on the controlled opening. We do agree that omitting such details did make our previous manuscript too simplistic.

Comments to the Author:

2. Herein the authors fabricated the nanostructure using fullerene. The reason of choosing fullerene rather than other materials should be addressed. Potential applications of these nanostructures utilizing the unique optoelectronic properties of fullerene should be discussed.

Response: We thank the reviewer for the valuable suggestions.

C₆₀ is the first system where the new synthetic control is established. It is a popular material used for optical devices and solar cell. Among all molecular nanocrystals and nanoparticles, C₆₀ is probably the one with richest morphologies. Like Au nanostructures are intensely studied as the model system for metal nanostructures, C₆₀ is often taken as the model system for molecular nanostructures.

In addition to C₆₀, C₇₀ and PCBM (phenyl-C₆₁-butyric acid methyl ester) can also give similar hollow structures. We have now included the C₇₀ and PCBM data in our revised manuscript (Supplementary Fig. S18).

In terms of the potential applications of these structures utilizing the unique properties of fullerene, we have added a corresponding discussion. We hope this synthetic advance could inspire new potentials for future applications.

Reviewer3:

Comments to the Author:

Fei Han et al. reported a very interesting method to synthesize hollow-structured fullerene. Hollow nanostructures have broad potential applications in nanoreactor, catalysis, energy storage, and so on. However, the fabrication of hollow nanostructures of tailored shapes on nano-scale remains significant challenges. In this work, the author reported a facile synthetic methodology of nanopottery, which could create different hollow units and connection of hollow units. The key innovation of this study is to control the shape of liquid droplet and then use them as template for a variety of hollow C₆₀ nanostructures. In fact, the liquid droplets were always spherical to minimize the surface tension in most system which was difficult to manipulate their shape. As such, the quality of the experimental data is superb and the manuscript is well written. The idea is novel and the experiments were well designed and conducted to prove the concept. If the fundamental mechanism and design principle could be more comprehensively understood, it would provide broader impact. I would recommend this work to be published if following concerns could be addressed:

Response:

We thank the reviewer for the high recognition of the significance, novelty, and the positive comments on our works.

To help in the comprehensive understanding of our mechanism, we provided detailed discussion on the mechanism in our revised manuscript. Also, more control experiments and discussions were added to verify the observations in intermediates, and to support the proposed mechanism. We do agree that the mechanism discussion in our previous manuscript could be improved.

Response to the reviewer's questions/concerns:

1. All the nanostructures created with nanopottery were characterized by TEM. Could the author provide more characterizations with statistics significance? For example, the dynamic light scattering (DLS) can not only provide statistical information in the size but also in the structure factor of different hollow structures. How about the size and structure changes of nanobowls and nanobottles with various nodes or junctions?

Response: In terms of the characterizations with statistics significance, we did DLS measurements for different hollow structures to reflect their overall shape changes. We also measured the structural parameters of different hollow structures, which cannot be characterized by DLS or in solution phase. For example, the inner diameter and the opening size of nanobowls; the neck length and the shell thickness of nanobottles (**Page 2**).

In multi-nodal structures (as mentioned in **Page 3, line 27**), with the addition of nodes, size of original nanobowls/bottles slightly increased due to the deposition of C₆₀ onto their outer surface. Such changes could be readily measured from the TEM images. However, it would be difficult to directly measure these structural features in solution phase, as the multiple nodes are connected

as a whole. In our revised manuscript, to help researchers know more about their structure information, we fitted the proportional changes between the volume of the C₆₀ solution and the cavity size in cucurbit growth with a linear function.

2. What is the driving force for the C₆₀ in xylene to aggregate together and form the shell of the structure?

Response: The driving force leads to the aggregation/deposition of C₆₀ is the reducing of its solubility and the intermolecular force. As mentioned in **Page 2, line 8**, IPA, a poor solvent of C₆₀ was added to reduce its solubility.

3. The authors only mentioned the influence of the rate of xylene on the structure. How about the influence of the rate of IPA?

Response: The major role of IPA is to reduce the solubility of C₆₀. Hence, so long as IPA was not added too fast (*e.g.* in one-shot), it will not influence the formation of nanobowls.

4. Could the authors explain how the fullerene assembles into different structures? What is the binding morphology between the C₆₀?

Response: The C₆₀ aggregated due to the reducing of solubility. Just like common C₆₀ crystals, we believe C₆₀ molecules in these hollow structures were mainly bound via Van der Waals' interactions, plus some π - π stacking and hydrophobic interactions (*New J. Chem.*, 2008, **32**, 159-165; and *J. Mol. Struct-Theochem.*, 2003, **625**, 189-197).

5. According to the HR-TEM and SAED results, the C₆₀ shell in these nanobowls is amorphous. Are these nanocontainers durable in all pH environments? Can they stand heating? How about the chemical stability of nanobowls or nanobottles with various nodes or junctions?

Response: Thanks for the valuable suggestion. In our revised manuscript, we have added more stability information. As described in **Page 2, line 24** and **Page 7, line 29**, they are stable when transferred into water, and cannot be damaged by vigorous sonication. Both nanobowls and hollow structures with multiple nodes can bear a mild heat treatment at 80 °C for at least 24 h.

6. The Figure 4F in the supplementary is missing.

Response: Thanks for pointing out our mistake, Supplementary Figure 4F should be **4D**, we have

already corrected this in the revised manuscript.

7. From the SEM images showed in Figure 2 and 3, the surface of the nanoparticle is not smooth. What is the reason for this?

Response: Thanks for the in-detail observation of our SEM images, but we already mentioned this in the **note of Supplementary Figure S3**, the rough surface should come from the gold layer coated on the sample before SEM analysis (as described in **Characterization** part).

8. The authors used xylene, DMF and IPA as the solvents for all the experiments. How about other organic solvents, such as ethanol, methanol, THF, CHCl₃, acetone and DMSO?

Response: Thanks for the valuable reminding. To elaborate this point, we added the data of **generality** in our revised manuscript. In short, method of nanopottery can be applied to other solvents, *e.g.*, within those you mentioned, ethanol and methanol; other organic solvents are possible as well.

9. In Figure 4i, the authors showed the connection of the hollow units. Is it possible to separate the dimer and trimer?

Response: As the size of dimer and trimers is too close, it is hard to separate them with a simple method. However, the monomers and dimers could be separated via centrifugation.

10. To broader application of this method, how to remove the excess C₆₀ in the solution after the synthesis? In the present manuscript, the authors did not mention the purification process. Also, is the product with different nanostructures stable and easy to store? The authors are invited to add some more detailed information about this in the revised manuscript.

Response: Thanks for the valuable suggestions. The purification process have been mentioned in our manuscript, **Characterization** part, and the remaining explanations were listed as follow:

Purification: Because the excess C₆₀ stays oversaturated in DMF/IPA phase (that is, out of the nanobowl), we could simply separate these nanobowls from the solution with centrifugation without any further purifications, just like the separation of nanocrystals from the solution phase.

Storage: We have added more details in our revised manuscript. As mentioned in **Page 2, line 24** and **Page 7, line 30**, dimers of these hollow structures can be stored in original sample solution for at least 3 months. Individual hollow structures (bowls, bottles, spheres) have even longer storage time.

11. How to apply this material is an important issue. Could the authors add experiments to illustrate some potential application, for example, the possibility of application the nanoparticles contained in the nanobottles.

Response: In terms of potential application, we construct the multi-compartment nano-containers as a proof-of-concept. Through this synthesis and understanding the underlying mechanism, we want to demonstrate that nanoparticles could be selectively incorporated, as an extension of the traditional nanocontainer concept.

Indeed, in nanoscience, it remains a great challenge to demonstrate the usefulness of sophisticated structures. We attempted to give an example of how to use the vessels to holding and isolating different nanoparticles. Such site-selective incorporation is a synthetic advance, as well as a fundamental step towards future applications. We have added a discussion regarding to the potential applications of these structures. We hope this synthetic advance could inspire new potentials for future applications.

REVIEWERS' COMMENTS:

Reviewer #1 (Remarks to the Author):

The authors have addressed my comments adequately.

Reviewer #2 (Remarks to the Author):

The authors have properly addressed all the issues from the referees. The revisions made to the manuscript are satisfactory.

Reviewer #3 (Remarks to the Author):

The authors have carefully addressed the questions and revised the manuscript. The current experiments were well-done and the manuscript was well written after the revision. Therefore, I recommend this paper to be published on Nature Communications.